# Method for Isolation of Myxozoan Proliferative Stages from Fish at High Yield and Purity: An Essential Prerequisite for In Vitro, In Vivo and Genomics-Based Research Developments

**DOI:** 10.3390/cells11030377

**Published:** 2022-01-23

**Authors:** Ana Born-Torrijos, Anush Kosakyan, Sneha Patra, Joana Pimentel-Santos, Brian Panicucci, Justin Tze Ho Chan, Tomáš Korytář, Astrid S. Holzer

**Affiliations:** 1Institute of Parasitology, Biology Centre of the Czech Academy of Sciences, 37005 České Budějovice, Czech Republic; borntorrijos.ana@gmail.com (A.B.-T.); anna.kosakyan@gmail.com (A.K.); snehampatra@gmail.com (S.P.); joanapimentelsc@gmail.com (J.P.-S.); bpanicucci@paru.cas.cz (B.P.); justin.chan@paru.cas.cz (J.T.H.C.); tkorytar@paru.cas.cz (T.K.); 2Laboratory of Ecological Plant Physiology, Global Change Research Institute of the Czech Academy of Sciences, 60300 Brno, Czech Republic; 3South Bohemian Research Center of Aquaculture and Biodiversity of Hydrocenoses, Faculty of Fisheries and Protection of Waters, University of South Bohemia, 37005 České Budějovice, Czech Republic

**Keywords:** diethylaminoethyl (DEAE) cellulose, cell separation, cytometry, anti-carp antibody, *Sphaerospora*, parasite, blood stages, teleost, common carp

## Abstract

Myxozoans are a diverse group of microscopic cnidarian parasites and some representatives are associated with important diseases in fish, in both marine and freshwater aquaculture systems. Research on myxozoans has been largely hampered by the inability to isolate myxozoan parasites from their host tissues. In this study, we developed and optimized a method to isolate the myxozoan proliferative stages of different size and cellularity from fish blood, using DEAE-cellulose ion exchange chromatography. We optimized several parameters and obtained 99–100% parasite purity, as well as high survival and infectivity. Using polyclonal pan-carp blood cell-specific antibodies, we further developed a rapid cytometric assay for quantification of the proliferative stages, not only in highly concentrated DEAE-C isolates but also in dilute conditions in full blood. Early developmental stages of myxozoans are key to parasite proliferation, establishment, and pathology in their hosts. The isolation of these stages not only opens new possibilities for in vivo and in vitro studies, but also for obtaining purified DNA and protein extracts for downstream analyses. Hence, we provide a long-desired tool that will advance the functional research into the mechanisms of host exploitation and immune stimulation/evasion in this group, which could contribute greatly to the development of therapeutic strategies against myxozoans.

## 1. Introduction

Myxozoa is a large and diverse group of microscopic endoparasites belonging to the phylum Cnidaria. Myxozoans are characterized by a two-host life cycle, alternating between invertebrate (mostly annelid) and vertebrate hosts (mostly fishes). They are especially known for the diseases they cause in wild and cultured fishes. Myxozoan outbreaks are of particular importance in the light of climate change, which affects myxozoan geographic distribution and disease severity, leading to new emergence scenarios [1,2,3]. It is of concern that, at present, no licensed treatments or vaccines exist against myxozoans in fish, that are effective but also guarantee the safety of products for human consumption.

For the development of targeted antiparasitic strategies, research methodologies that can better characterize myxozoans and their host-parasite interactions at the cellular and molecular levels, are urgently required. Since the publication of the first genome [4], myxozoan ‘omics’ has become a rapidly progressing field and primary source for an improved understanding of myxozoan host exploitation mechanisms; however, contamination from hosts and environmental organisms pose major obstacles for the assembly and analysis of high throughput sequencing datasets [5], which is further complicated by the extremely derived character of myxozoan genomes.

Contamination of parasite isolates by host cells also poses problems in experimental infections where timing, dosage and route are variables that need to be strictly controlled for reproducibility and validity. Protocols for parasite purification are crucial for the decontamination of genomic datasets, and to study myxozoan–fish host interactions in vivo and in vitro in host- and pathogen-free culture systems; however, the number of successfully established protocols is extremely limited for this group of parasites.

Both sucrose [6] and Percoll [7] density centrifugation have been successfully used to isolate the spores of *Myxobolus* spp. and *Kudoa thyrsites* from host tissues. Based on the differences between the surface structure of spores and host cells, spores were also successfully isolated using an aqueous bi-phase system of dextran and polyethylene glycol [8,9]. Nematocysts of myxozoan spores were purified by sonication followed by Percoll centrifugation [6] and by using a tailored dielectrophoresis-based microfluidic chip [10], which allowed for the analysis of the nematocyst protein repertoire. Due to their distinct physical properties, i.e., hard spore valves and differential density, the isolation of spores is easier than that of the parasite proliferative stages whose cell surface differs little from that of the surrounding host cells. With growing interest in myxozoan invasion, migration, proliferation, and interaction with their hosts, a need for the physical isolation of pre-sporogonic developmental stages emerged. Pre-spore stages were successfully concentrated by density centrifugation [11,12] but they have never been isolated completely from host cells. Hence, the development of new physical separation methods is urgently required in this group of parasites.

Our laboratory model organism, *Sphaerospora molnari*, is a myxozoan parasite of common carp (*Cyprinus carpio*) and it is the causative agent of gill, skin and blood sphaerosporosis [13,14]. Similar to other true sphaerosporids in their respective hosts [15,16,17,18], *S. molnari* forms fast proliferating, pre-sporogonic stages in the blood of common carp [19]. These stages, known as blood stages (BS), are easily recognizable due to their motility (see videos in [19]). When blood is centrifuged in hematocrit capillary tubes, these stages co-localize with leukocytes in the “buffy coat” located on top of the denser erythrocytes [20]. Following the same principle, larger volumes can be processed using Ficoll-Paque [12,20]; however, while strongly concentrated in the buffy coat, the parasites are not completely isolated from the host cells. This is problematic for in vitro cell culture assays and for in vivo trials, where the co-injection non-autologous leukocytes will alter immune responses.

Isolation of one or multiple cell types from a heterogeneous population is an integral part of modern biological research and is essential for basic cell biology research [21]. The central principle of separating any cell type from a population is to select for properties that are unique to that cell type. The most widely used cell isolation and separation techniques can be broadly classified based on (1) surface charge and adherence, (2) cell size and density, (3) cell morphology and physiology, or surface markers (often targeted by antibodies). In our quest for the development of an isolation method for the myxozoan proliferative stages from blood, we ensured that the parasites would endure minimal mechanical, chemical and physiological stress to maintain the integrity of their typical cell-in-cell structure, while achieving a high parasite purity and yield. Due to the lack of characterized *S. molnari* surface antigens for antibody-based methods, and the morphological diversity of stages with high variability in cellularity and size [12], we opted for a method based on surface charge or adhesion.

Half a century ago, Lanham [22] discovered a technique to separate trypanosomes, agents of human trypanosomiasis or sleeping sickness, from the blood of infected rodents, based on anion exchange chromatography using diethylaminoethyl cellulose (hereafter DEAE-C). Between pH 6–9, the surface of trypanosome blood parasites is less negatively charged than that of the blood cells of the mammalian host. When running blood through a DEAE-C column, the blood cells are retained, whereas parasites are eluted through the column. To date, this method is superior to all other available techniques for trypanosome detection and is indispensable for the production of antigen for agglutination tests and other assays [23]. After initial separation trials with *S. molnari*, we realized that a similar charge difference exists between myxozoan BS and fish blood cells. We adapted, developed and optimized the DEAE-C anion exchange chromatography for BS separation, maximizing the parasite yield and purity by varying several conditions (column height, elution buffer composition, ionic strength) and analyzing cellularity, viability and infectivity of the parasites prior to and after elution. Finally, based on host cell antibody staining, we developed a rapid flow cytometric method for the quantification of parasites in full blood and in DEAE-C isolates. We analyzed data in a comparative approach to provide optimized methods for microscopic and flowcytometric analyses. To the best of our knowledge this is the only protocol that has achieved complete isolation of the myxozoan proliferative stages from fish, and it opens an array of new possibilities for future in vitro, in vivo and genomic research in myxozoans.

## 2. Materials and Methods

### 2.1. S. molnari BS Collection and Ficoll Isolation

*S. molnari* blood stages were collected from a long-term in vivo culture line, laboratory-maintained for 4+ years in specific pathogen-free (SPF) carp, in which BS have been transferred from fish to fish by intraperitoneal (IP) injection following established protocols [20]. Briefly, infected fish were maintained at 21 °C and were bled 4–5 weeks after IP injection of BS. Full blood was collected from the caudal vein of the fish using heparinized syringes, and immediately mixed with the same volume of RPMI 1640 medium (Life Technologies Limited, Paisley, UK, hereafter RPMI), inverting the tube several times to avoid coagulation. The BS and leukocytes were co-isolated using Ficoll-Paque (Cythra, Uppsala, Sweden, hereafter Ficoll). A maximum of 3 mL of the blood-RPMI solution was layered onto 2 mL of Ficoll and centrifuged for 20 min at 800× *g*, with gentle acceleration and braking. Lymphocytes and BS localize together in the interphase. Cells were collected with a pipette and were washed twice with RPMI (700× *g*, 5 min) to remove excess Ficoll and plasma components. To reduce animal usage and comply with the 3Rs of animal welfare (replacement, reduction and refinement), we increased the parasite yield from infected fish by immunosuppression, using triamcinolone acetonide at a dose of 200 µg/g of body weight, the day before parasite IP injection [24]. Following anesthesia with 0.6 mL/L clove oil, we bled out 14 fish (total length 12.5–16 cm) for the different comparative approaches performed in the present study.

### 2.2. DEAE-C Setup and Variation of Column Parameters

To optimize the DEAE-C-based BS separation method, we tested elution buffers (i) of different ionic composition, and (ii) of different ionic strength, as well as (iii) different column lengths. All assays were compared to Ficoll isolations (our standard method).

i.In the original protocol isolating trypanosomes from a variety of vertebrates, ‘phosphate-saline-glucose’ was used to move the cells through the DEAE-C column [25]. Similar to this setup, we used PBS (phosphate-buffered saline; ionic strength 0.164 M, pH 7.4) supplemented with 1.5% glucose. This was compared to commercial RPMI, our standard host cell culture medium (containing 2% glucose, ionic strength 0.152 M, pH 8.0) and a custom salt solution we prepared, which was of the same ionic strength and composition as RPMI but lacking amino acids, vitamins and other non-salt components, which we hypothesized were not necessary for cell separation (hereafter RPMIsalt; composition: 100 mg/L Ca(NO_3_)_2_ 4H_2_O, 100 mg/L MgSO_4_ 7H_2_O, 400 mg/L KCl, 6000 mg/L NaCl, 800 mg/L Na_2_HPO_4_ [anhydrous], 2000 mg/L NaHCO_3_ and 2000 mg/L D-glucose, in Millipore water; ionic strength 0.152 M, pH 8.3–8.4). RPMIsalt was sterilized by filtration (0.02 μm filter).ii.To test the effect of the column height (length of packed DEAE-C on a given diameter of the column container) on the efficiency of parasite elution, we compared packed DEAE-C volumes of 15 mL and 7.5 mL in 1.5 cm × 12 cm polypropylene chromatography column containers (Econo-Pac^®^, Bio-Rad, Hercules, CA, USA). Due to the better yield of shorter columns, we further reduced the column height to only 5 mL volume, the minimum volume that still ensures the retention of host cells. To optimize the processing of larger amounts of whole blood and as an alternative to preparing multiple DEAE-C columns, we tested if the stages first separated from host red blood cells by Ficoll could thereafter be separated from host leukocytes by DEAE-C.iii.Ionic strength is key to the separation of the parasites from the differently charged host erythrocytes. At low ionic strengths, competition for charged groups on the ion exchanger is at a minimum and substances are bound strongly. Increasing the ionic strength increases competition and reduces the interaction between the ion exchanger and the sample substances, resulting in their elution. The surface charge on erythrocytes varies with species [26], and so does that of trypanosomatids [27]. Hence the isolation of different trypanosome species from their respective hosts requires elution under different ionic conditions [25]. To optimize separation and hence increase parasite purity, we compared the isolation protocols using four ionic strengths of 0.076 M (0.5×), 0.152 M (1×), 0.1823 M (1.2×) and 0.228 M (1.5×) RPMIsalt.

DEAE-C columns were prepared by adding 3 g of DE52 cellulose (Biophoretics, Sparks, NV, USA; granule size 25–60 μm, exchange capacity 0.9–1.4 mmol/g, water content 65–75%, maximum flow rate 50 mL/min, protein loading capacity 550–900 at pH 8.5) to 100 mL of medium (PBS/RPMI/RPMIsalt). The resulting slurry was mixed with a magnet stirrer and then left to settle for 10 min. To discard the smallest cellulose fibers, the slurry was decanted three times before fresh medium was added to resuspend the cellulose. The plugged chromatography columns were fixed vertically on holders. The slurry was carefully poured into the column and left to settle. Once settled, medium was gently added to the top with a syringe, without disturbing the surface of the settled fibers. Upon removal of the plug, additional medium (approx. 15 mL) was allowed to flow through the column and compact the fibers. For BS isolation from the fish blood, 0.5–2 mL of fresh, non-coagulated blood was carefully pipetted onto the surface of the plugged cellulose, without disturbing/resuspending it. Next, we resumed flow of the column to allow for cell binding by adhesion or elution by gravity. A total volume of 100 mL was used for elution of the BS. All eluates were centrifuged (800× *g*, 8 min) and the pellets were resuspended in a known volume of RPMI to estimate the total number of BS (Bürker chamber or flow cytometry, see below). At no point throughout the preparation and parasite isolation was the DEAE-C allowed to dry.

### 2.3. Evaluation of Parasite Yield, Purity, Integrity, Viability and Infectivity

To estimate BS recovery after isolation, their initial numbers in whole blood had to be determined. For microscopic counts, aliquots of whole blood were centrifuged in heparinized capillary tubes (75 mm/75 μL; Hirschmann Laborgeräte GmbH & Co. KG, Eberstadt, Germany) at 4000× *g*, for 5 min. After breaking the capillary tubes, the layer containing a mix of leukocytes and BS was collected (including the top layer of erythrocytes), using a micropipette. The collected cells were diluted in 400 μL of RPMI. All samples were quantified in triplicate in a Bürker chamber, where BS in the three big diagonal squares were counted, under the microscope (400×; Olympus BH2, Düsseldorf, Germany). The BS were counted unstained as they can easily be recognized by their motility (supplementary videos in [19]), non-motile cells were counted as host cells.

To determine the effect of the isolation method on the viability of the parasites, the isolates were stained with 10 μg/mL fluorescein diacetate (Thermo Fisher Scientific, Prague, Czech Republic; life stain) and with 1 µg/mL propidium iodine (Thermo Fisher Scientific, Prague, Czech Republic; dead cells) in the dark, at room temperature, for 10 min. Thereafter, 3 × 100 cells per isolate were checked for BS viability by fluorescence microscopy (Olympus BX51 microscope; 1000× magnification).

To compare the cellularity of the BS prior to and post isolation, the total number of nuclei present in each BS was determined, which in *S. molnari* equaled the number of cells they are composed of [19]. BS isolates were dropped onto grease-free glass slides, left to settle for 5 min, fixed with methanol for 1 min, before being dried and mounted in Fluoroshield containing DAPI (Sigma-Aldrich, Prague, Czech Republic). The number of nuclei in 200 BS per fish was counted on a fluorescent microscope (see above). The presence of host cells (leukocytes, thrombocytes, erythrocytes) was also noted.

To determine the infectivity of the BS isolated from each isolate, carp fry of 6–8 g were injected with 10,000 BS per g of body weight.

### 2.4. From Large to Small: An Attempt to Separate Parasite Stages of Different Size/Cellularity

Myxozoans are miniature, pluricellular organisms with a characteristic cell-in-cell arrangement where a primary cell holds one or more secondary cells which themselves can hold tertiary cells within their cytoplasm, resulting in different cellularities of individual parasites. Since each cell type may have different characteristics [28] we tested the possibility of separating different developmental stages of the BS. We hypothesized that smaller BS can be separated from larger BS due to the size of their surface area and hence adhesion to the DEAE-C column. Fourteen 5 mL fractions were sequentially collected throughout the elution of parasites, originating from a blood sample containing a high morphological variety of BS.

### 2.5. Development of an Alternative Method for Parasite Quantification by Flow Cytometry

To speed up cell counts and to develop a more accurate protocol for strongly diluted parasite samples, such as infected whole blood, we devised a method using flow cytometry and compared the results with those obtained by microscopy. We chose a negative labelling approach (labelling host cells preferentially) due to a considerable lack of knowledge about *S. molnari* surface proteins and to avoid interference with parasite viability for downstream applications.

We produced mouse polyclonal anti-pan-carp blood cell antibodies by immunizing BALB/c mice with 10 million carp red blood cells and IgM+ splenocytes, via IP injection. The mice were boosted two times at two-week intervals before a final immunization four days prior to spleen harvest. To prepare the cells, the spleens of SPF carp were homogenized and passed through 100 μm cell strainers (Corning Incorporated, Durham, NC, USA), before density centrifugation using 25% Percoll (GE Healthcare, Uppsala, Sweden) in RPMI. Red blood cells were isolated by Ficoll centrifugation (see Section 2.1). Pelleted cells were collected and excess Percoll/Ficoll washed off with RPMI. The splenocytes were stained with WCI 12, a mouse monoclonal antibody specific for carp IgM [29] and magnetically selected using LS columns (Miltenyi Biotec, Bergisch Gladbach, Germany), according to the manufacturer’s instructions.

Hybridoma fusions were performed as described by Köllner et al. [30], with the exception that the SP2/0-Ag14 cell line was cultured in an IMDM medium (Life Technologies Limited, Paisley, UK), supplemented with 10% fetal bovine serum and GlutaMAX (Life Technologies Limited, Paisley, UK). Primary mouse peritoneal macrophages were collected by peritoneal lavage and plated one day prior to fusion. These cells plus fused hybridoma cells were cultured under selection with a HAT supplement (Life Technologies Limited, Scotland, UK). The antibodies were purified from the hybridoma supernatant using the Mouse TCS Antibody Purification Kit (Abcam, Cambridge, UK), according to the manufacturer’s instructions. The isolated pan-carp blood cell-specific antibodies were detected with either secondary goat anti-mouse IgG-Alexa Fluor 488/647 antibodies (Invitrogen, Rockford, IL, USA) or directly labeled with either the APC or R-PE Conjugation/Labeling Kit (Abcam, Cambridge, UK) and preserved in 0.05% sodium azide. 

We first validated the anti-pan-carp blood cell antibody reagent for staining of the host cells but not the parasites. Having confirmed the specificity of the reagent, we then stained mixed populations of parasites and host cells from infected fish blood and BS isolates from the Ficoll or DEAE-C columns. For this procedure, isolated cells or whole blood (1 μL) washed twice in RPMI were stained with approximately 1 μg of anti-pan-carp blood cell-APC/R-PE in a volume of 200 μL of RPMI, for 15 min. Cells were washed with and resuspended in 200 μL of RPMI for flow cytometric analyses on a BD FACSCanto II (BD Biosciences, Prague, Czech Republic). The number of parasites per μL was enumerated based on records of analysis time, flow speed and the original volume.

The results from flow cytometry and microscopy were compared by performing a correlation, using data from the different isolation methods (Spearman rank correlation; R Core Team, version 3.6.0).

All methodological comparisons were performed in three to nine replicates (full details given in relevant tables and figures).

## 3. Results

Using optimized DEAE-C parameters, we were able to isolate *S. molnari* BS from the blood of carp, with an average of 46% and a maximum of 68% recovery of parasites initially counted in infected whole blood, allowing us to isolate up to 28,000,000 BS per ml of blood. While we achieved a higher yield with our standard density centrifugation method, Ficoll (average yield of 72%), the BS concentrated by Ficoll remained highly contaminated with host cells and did not represent the whole morphological variety of stages present in the blood. In contrast, the optimized DEAE-C method isolated *S. molnari* stages with over 99–100% purity and includes all parasite sizes.

### 3.1. Optimized Conditions for S. molnari BS Recovery from DEAE-C

Column height strongly affected the *S. molnari* BS recovery (Figure 1), with longer columns (15 mL) retaining up to four times more parasites (5.5% to 22.8% yield) than shorter ones (7 mL and 5 mL; 19.8–46.3% yield). Reduction of the DEAE-C column height from 7 mL to 5 mL resulted in an average additional 4.8% parasite recovery. Thereby, a column of 5 mL ensures the retention of host blood cells, even when the general blood parameters are greatly affected by parasitemia and the number of erythroblasts is extremely high [24]. Further reduction of the column height can lead to the release of erythrocytes and especially erythroblasts from the column into the eluate. Using Ficoll prior to DEAE-C resulted in substantial parasite loss in the 15 mL columns (only 1.4% average yield) and a minor reduction in the 5 mL columns (32.1% average yield). Independent from the column height, when the Ficoll isolations were performed prior to DEAE-C, we observed a negative effect on parasite morphology (irregular surface structure) and motility (slower movement).

We found that the elution buffer strongly influenced the adsorption-elution of *S. molnari* to and from the column (Figure 1). PBS was not suitable as an elution buffer and achieved the lowest yields, averaging 5.4% at a 15 mL column height and 19.8% at a 7 mL column height. In addition, we noticed that parasites eluted in PBS showed changes in morphology and motility similar to the changes after the combined Ficoll/DEAE-C isolation. RPMI was a superior elution buffer with an average of 22.8% and 29.6% parasite yield at 15 mL column height and 7 mL column height, respectively. The more economical RPMIsalt performed best with an average of 41.5% BS yield at a 7 mL column height and 46.3% at a 5 mL column height.

Low ionic strength of the RPMIsalt (0.076 M, 0.5×) was shown to retain most parasites while 0.152 M (1×) RPMIsalt recovered 35.2–55.2% of parasites, without host cell contamination. At a 0.228 M (1.5×) ionic strength, 63–86.3% of the BS were recovered but at the cost of host cells making up most of the eluate. BS recovery at a 0.183 M ionic strength (1.2× RPMIsalt) was 5.4% higher than at 0.152 M (1× RPMIsalt) but demonstrated a minor contamination with erythrocytes; however, we achieved a 62% pure BS recovery if we eluted with 70 mL of 0.152 M (1×) RPMIsalt, followed by 30 mL of 0.183 M (1.2×) RPMIsalt.

### 3.2. Vitality, Integrity and Infectivity of S. molnari BS Post DEAE-C Isolation

Mortality of the DEAE-C-isolated BS was between 0% and 6%. The optimized method using RPMIsalt and a low column height resulted in an average mortality of only 0.72% (0–1.7%). Most importantly, the DEAE-C method appeared to elute all parasite stages, independent of their cellularity (2 to 20 cells; Figure 2, Table 1), hence preserving the natural variability of parasite sizes and developmental stages. Our data indicates that an average of 74% (63–87.8%; Figure 2a, Table 1) of parasite stages are naturally bicellular in whole blood. These cell doublets represent a primary cell enveloping a secondary cell, while a smaller percentage is pluricellular; however, parasite cellularity in the isolates from different fish can vary considerably (Table 1), likely representing different parasite developmental stages. We found Ficoll separation to recover predominantly stages of 2–4 cells, while a substantial number of larger stages was lost (Table 1) because they are more dense and able to pass through the Ficoll layer, pelleting with the erythrocytes (personal observation in pellets). In the PBS columns, large parasites were found to break up and release small cell doublets, thereby increasing the relative percentage of these stages to an average of 91.4% (Figure 2, Table 1). PBS eluates from the large DEAE-C columns were more negatively affected than those of the shorter ones, and they completely lacked parasite stages >8 cells, while shorter columns were found to recover stages up to 13 cells. RPMIsalt eluates from the DEAE-C columns retained the natural variety of stages almost completely with an average of 74.8% (64.2–81.2%) of parasites being cell doublets and even the largest stages (20 cells) being eluted without rupture of the primary cell membrane. Thereby, the cellularity of the isolated parasites strongly reflected that of the original population in full blood.

All DEAE-C- and Ficoll-isolated parasite populations were able to infect and propagate in fish, with parasitemia levels in the recipient fish peaking between 4 and 6 weeks post infection.

### 3.3. Parasite Elution by Fraction Allows Partial Separation According to Size

When using the DEAE-C columns for parasite separation, the largest bulk of parasites was recovered in the first four fractions of the eluate (67.6% in the first 20 mL; Figure 3a) while parasite recovery was stable until fraction 11 (94.1% of total; 55 mL; Figure 3a). Thereafter parasite elution continues at a low level (1–2% per 5 mL), possibly beyond the 14 fractions obtained in this study.

The first three fractions showed a steady increase in the average cellularity of parasite stages eluted (Figure 3b). The largest stages were eluted in fractions 4–6, followed by an abrupt decrease and a dominance of cell doublets (82.5–95% in fractions 4–6 versus 51–64.5% in fractions 1–6; Figure 3b).

### 3.4. Flow Cytometry as a Semi-Automated Alternative Method for Enumerating BS

To count the parasites in untouched whole blood and determine host-contamination in a larger population of BS isolates, we developed a flow cytometry protocol. Due to the lack of suitable (commercial) reagents reactive to both carp blood cell populations and the parasite, we successfully produced and purified a polyclonal pan-carp blood cell-specific antibody reagent and developed a negative antibody staining approach.

When testing the pan-carp blood cell-specific antibody reagent on the whole blood of non-infected carp, an over seven-fold change in mean fluorescence intensity allowed us to correctly identify 100% of host cells (Appendix A). The antibody preparation was then directly labeled with fluorophores to generate a reagent for a one-step staining protocol. The conjugated antibody maintained specificity to host cells (Figure 4a) while the staining of DEAE-C-purified parasite was weak and incomparable to that of the host cells (Appendix A).

Using fitted gates specific to each individual and with the parameters being analyzed, we were able to confidently exclude host cells in the analyses of whole blood (Figure 4b, bottom row) while determining that DEAE-C parasite isolates were >99% pure (Figure 4b, top row), which was estimated at 100% purity in the microscopic counts. Alternate gating strategies were applied to further enhance the resolution and accurate parasite identification (Appendix A). Size and complexity/density profiles of the DEAE-C-purified parasites and that of parasites from whole blood were comparable (Figure 4b), hence validating the flow cytometry method for the BS of different cellularity/complexity, as determined by microscopy.

We tested the flow cytometric identification method at different steps throughout the DEAE-C purification process to quantify the parasites in parallel with microscopy. The correlation between the microscopy and flow cytometry counts was highly significant (rs = 0.97, *p* < 0.001, *n* = 12; Appendix A), hence confirming the validity of the new flow-cytometric application.

## 4. Discussion

Due to the lack of known myxozoan antigens and the cryptic nature and relatively large size variation of the myxozoan developmental stages (in the case of *S. molnari* 2.4–12 μm in diameter), charge differences between host and parasite cells likely represent a unique differential characteristic allowing host–parasite separation by anion exchange columns, such as DEAE-C. Unsurprisingly, 50 years after its discovery and first application for trypanosomes in mammals [25], the same method remains the gold standard for isolation of these protists [23]. The method is based on surface charge, with retention of more negatively charged cells, in this case erythrocytes and leukocytes, in the DEAE-C column, while liberating less negatively charged ones, the parasites. This gives the method several advantages over other techniques which are density-dependent (e.g., Percoll, Ficoll, sucrose) and which we previously used as a standard for *S. molnari* BS concentration. Density-dependent approaches work well for uniformly sized and shaped cell fractions, such as myxosporean spores [31]; however, for *S. molnari* BS of varying cellularity we found that Ficoll only isolates a subset of small stages, alongside large numbers of host leukocytes that appear to have the same cell density. In contrast, we demonstrated that the cellular composition of *S. molnari* BS isolated by DEAE-C is highly diverse and very similar to that of the original parasite population in whole blood. This suggests that, given an adequate buffer, the anion exchange technique does not affect the physical integrity of the BS.

We noticed that the PBS-based elution buffer used for trypanosomes negatively affected the morphology and behavior of *S. molnari* BS. RPMI is often used for the isolation and cultivation of carp cells (e.g., [32,33]) and has been shown to be beneficial for *S. molnari*-based cell assays and morphology studies (e.g., [12,19]), likely because it is composed of a higher variety of cations when compared with PBS; however, RPMI is considerably more costly, as it contains vitamins, amino acids and antioxidants. In this study, we used RPMIsalt, an economical alternative to RPMI culture medium which has the same ionic composition, and hence the same properties that are relevant to the DEAE-C isolation process.

We adapted and further developed the DEAE-C method to achieve complete isolation, maximum yield and survival of the pluricellular myxozoan proliferative stages of *S. molnari* from fish blood. We found 1× RPMIsalt at 0.152 M ionic strength to be optimal but were able to elute more parasite stages when using 0.183 M RPMIsalt (1.2× RPMIsalt) in a second step. The combination of these salinities, the optimized RPMIsalt buffer and a minimal column length (5 mL) allowed us to elute an average of 46% and maximum of 68% of *S. molnari* BS from whole blood, at >99% purity. We found different isolates of *S. molnari* BS to be highly variable with regard to their cellular composition (compare F10, F11 and F12 in Table 1) and yield after isolation, and we were unable to obtain a higher yield. This may be partially explained by the fact that *S. molnari* BS feed on erythrocytes, and during this process attach to the host cells [24], hence likely being retained in the column due to their association with the differently charged and usually much larger host cells (the majority of BS represent cell doublets of 2.4–4 μm in diameter).

Given that populations of *S. molnari* BS of variable cellularity and size are eluted in different fractions, it is likely that the degree of their retention by DEAE-C is proportional to their negative surface charge, which changes with size. Thus, larger *S. molnari* stages composed of more cells appear to be retained for longer, due to their larger surface to volume ratio and a subsequent stronger adhesion to the column. We observed an increase in average cellularity over time/fraction. The abrupt decrease after fraction 6 to a low number of almost exclusively bicellular stages and their continuous subsequent elution in later fractions suggests that some large stages rupture in the column and liberate cell doublets. Despite cell doublets being present in all fractions, we were able to concentrate stages of higher cellularity in certain fractions of the DEAE-C column eluate. This approach in combination with single-cell/single-stage sequencing applications could well be used to try and identify the specific characteristics and functions of different cell populations in myxozoans, which have been the focus of much speculation but are so far based on morphological observations only (reviewed by Feist et al. [34]).

We successfully developed a flow cytometry assay for quantification of *S. molnari* BS which is unmatched regarding analytical speed and allows the processing of a comparatively large number of cells that reflects the overall cell population better. This is particularly important when enumerating parasite stages in full blood which is inherently difficult as red blood cells vastly outnumber the parasites, even when parasitemia levels are high. While we can resort to centrifuging whole blood in hematocrit capillary tubes and counting parasites in the mixed white blood cell/BS layer, bias is introduced when collecting the small volume of buffy coat by micropipette.

During the development of the flow cytometry method for *S. molnari* BS enumeration, we faced challenges with antibody sensitivity and specificity that required adjustments. We noticed an incomplete resolution of host cells and parasites, despite the application of a polyclonal antibody isolated from mice immunized uniquely with carp cells from carp that had never encountered the parasite. This problem may be related to the common amino acid sequence patterns of host epitopes and other surface-exposed non-epitopic residues [35], which are shared with *S. molnari*. The polyclonal nature of the antibody further increases the likelihood of cross-reactivity. On the other hand, antibody binding may well be specific to carp proteins but the parasites can integrate host cell surface proteins onto their own cell surfaces upon feeding on host erythrocytes (the most abundant cell population in the blood of the host) [24]. Therefore, this potential immune evasion strategy combined with the polyclonal nature of the antibody increases the likelihood of the parasite being stained. Conversely, the intensity of the fluorescence of cross-reactive antibodies on parasite cells was considerably lower than that of the host cells (Appendix A), and hence flow cytometer setting adjustments and gating strategies allow us to reliably differentiate the two; the intensity of specific staining of the host far outweighs that against the parasite, making the problem negligible. Any parasite binding by our reagent can potentially be mitigated by screening for the specific clones or excluding any antibody clones specific to the parasite using live parasite, lysed parasite, or derivatives. Much like how techniques such as chromatography are used to purify antibodies of interest, or how fetal bovine serum or bovine serum albumin are included in flow cytometry staining protocols, we may be able to increase the specificity of our polyclonal antibody without sacrificing sensitivity.

To further optimize the assay, we ensured that the parasite was undisturbed throughout the labelling process (negative labelling approach) and that parasite loss was reduced by a minimum number of steps. We achieved this by directly labeling the polyclonal antibody with a fluorochrome. Ultimately, we were able to measure suspensions of over 200,000 cells in just 20 s, following roughly 30 min of staining/washing/centrifugation, with reliable results showing strong correlation with counts from microscopy.

*S. molnari* shows a high proliferation rate in the blood [12] and the quantification of this developmental stage is hence of interest when testing parasite treatments or vaccination approaches ([36]). Our previous standard method for parasite quantification was qPCR which can detect much lower parasite quantities than flow cytometry due to the detection of amplified DNA, rather than the detection of cells; however, it is much more time consuming and cannot quantify exact parasite numbers when comparing, e.g., BS of *S. molnari* during early infections (21–28 dpi; 2–9 cells) or later ones (2–20 cells), due to varying amounts of nuclei/DNA per stage [12]. Then again, BS cellularity can only be determined by microscopy, as flow cytometry fails to estimate this parameter correctly, due to the miniature size of the parasites and the non-linear relation between fluorescence and nuclei of different sizes (personal observation; nuclei of primary cells are larger than those of secondary cells, etc.). A combination of different methods is hence necessary for an understanding of the specific proliferation processes of myxozoans in their hosts.

## 5. Conclusions

In summary, we have successfully established a physical isolation protocol for myxozoans that is fundamental for overall progress in the field. We predict that this protocol will find wide application in experimental investigations into cell and protein functions and the dynamics of protein distribution in myxozoans. This could provide valuable insights into the poorly known development of myxozoans and highlight molecular differences in their cell types. In our laboratory, DEAE-C anion exchange separation of *S. molnari* BS opened new avenues of research by enabling in vivo and in vitro assays, as well as the generation of host-free genomic and transcriptomic data, which is notoriously difficult to obtain in myxozoans [5,11]. Proliferating pre-sporogonic parasite stages have barely been characterized in the past [37] though they are key for understanding the parasites’ specific physiology, nutrition and host exploitation mechanisms during the early intrapiscine development. In the long run, the molecular characterization of these stages will be essential for designing specific antiparasitic strategies, especially for myxozoans important to the aquaculture sector.

## Figures and Tables

**Figure 1 cells-11-00377-f001:**
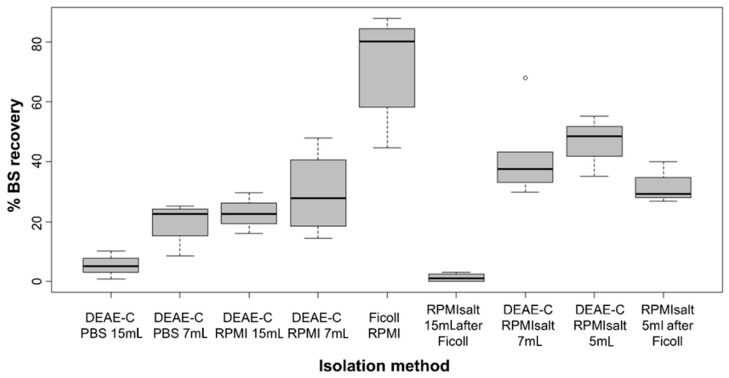
Comparison of *S. molnari* BS yields from different DEAE-C/Ficoll isolation protocols, using carp full blood as the starting material. Black horizontal lines represent the median, boxes represent 50% of the values, and upper and lower whiskers represent values >75th and <25th percentiles. Data based on blood samples of 3–8 fish each. ° represents an outlier.

**Figure 2 cells-11-00377-f002:**
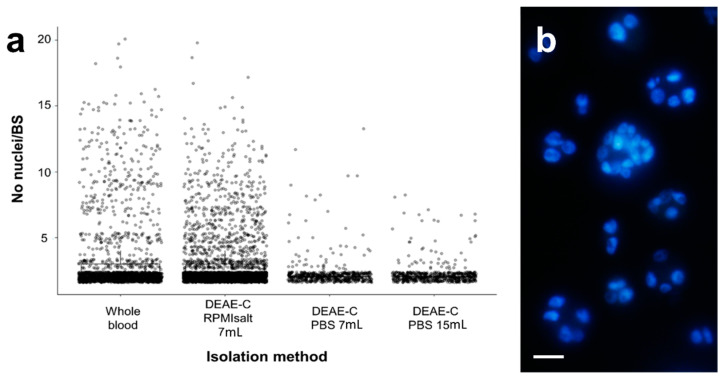
(**a**) Cellularity of *S. molnari* BS prior to isolation (in whole blood) and after isolation, using different methods (DEAE-C) and elution buffers. The corresponding numerical data is presented in Table 1. Grey circles represent jittered raw data of nuclei per BS counted in DAPI-stained smears. The total number of stages in each column differs depending on the number of replicates/fish (one replicate = 3 smears = 600 counted BS; Whole blood = 1800 stages (3 fish); DEAE-C RPMI salt 7 mL = 3000 stages (6 fish); DEAE-C PBS 7 mL and 15 mL = 600 stages (3 fish)). (**b**) DEAE-C isolated *S. molnari* BS, DAPI-stained to determine parasite cellularity. Image shows stages composed of 2–16 cells; size bar—5 μm.

**Figure 3 cells-11-00377-f003:**
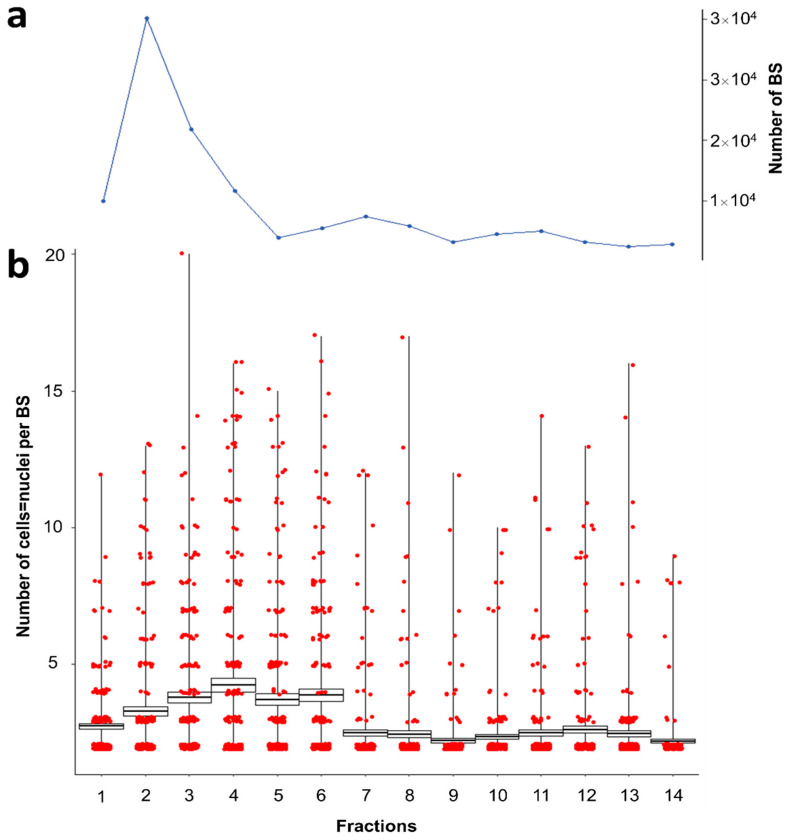
*S. molnari* BS sorting by size/cellularity: 14 fractions of 5 mL were collected from DEAE-C column (RPMIsalt, 5 mL column height). (**a**) Quantity of parasites per fraction. (**b**) Cellularity of *S. molnari* BS per fraction, estimated from number of DAPI-stained nuclei of 200 BS per fraction. Box plots represent the average number of cells/nuclei ± standard error (box) with maximum and minimum ranges (whiskers). Red circles represent jittered raw data from individual fractions.

**Figure 4 cells-11-00377-f004:**
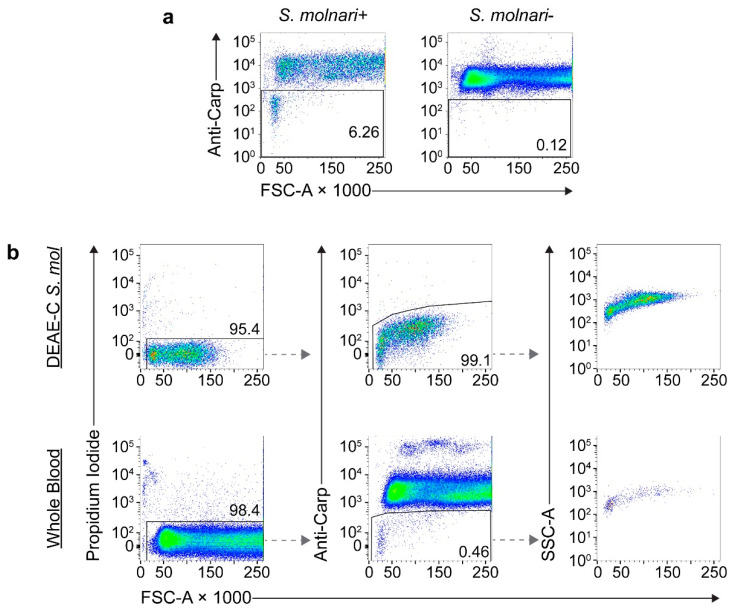
Flow cytometric approach for detecting and enumerating *S. molnari* BS using a negative labeling approach (all carp blood cells were labelled with the anti-pan-carp blood cell-APC/R-PE reagent [Anti-Carp]). (**a**) Quantification of BS in infected (+) vs. non-infected (−) whole blood. (**b**) Gating strategy for determining purity of *S. molnari* DEAE-C isolates (top row) and to quantify parasites in whole blood (bottom row). From left to right: First, only live cells (propidium iodide-negative) were gated (left); then, amongst these, parasites were detected by minimal reactivity to Anti-Carp (in this case 99.1% of the DEAE-C isolate and 0.46% of whole blood; middle); finally, we present side-scatter (SSC) and forward-scatter (FSC) profiles of BS isolates, that are comparable between pure parasite isolates from DEAE-C and in whole blood (right). Axes represent cell size (FSC), density/complexity (SSC) or fluorescence intensity (either of propidium iodide or of Anti-Carp staining). Dashed arrows illustrate step-wise gating and analyses of increasingly specific cell populations.

**Table 1 cells-11-00377-t001:** Cellularity of *S. molnari* blood stages in whole blood and after cell separation using different methods and elution buffers. The number of cells is equivalent to the number of nuclei, which was counted in 200 cells, after DAPI staining; F10–F12 represent samples from three individual fish that harbored parasite populations variable in composition and cellularity. Values in the table represent the percentage of the relevant stage in the whole parasite population.

Nr of Cells	Whole Blood			Ficoll RPMI	DEAE-C PBS 15 mL	DEAE-C PBS 7 mL	DEAE-C RPMIsalt 7 mL		
	F10	F11	F12	F12	F12	F10	F10	F11	F12
2	71.0	63.0	87.8	80.5	91.3	91.5	81.2	64.2	78.5
3	7.3	8.8	4.7	8.3	4.2	3.5	7.4	10.5	5.5
4	1.2	2.2	0.8	3.0	1.2	2.2	1.9	5.3	3.0
5	5.5	5.3	2.3	2.7	1.3	0.7	3.1	4.3	2.2
6	1.2	1.2	0.5	0.8	0.8	0.5	1.8	3.3	2.5
7	1.8	2.7	1.2	1.5	0.8	0.3	1.0	3.9	1.2
8	2.7	2.5	0.2	0.5	0.3	0.5	0.5	2.8	1.5
9	2.5	5.0	0.8	0.8	0	0.2	1.4	1.7	0.7
10	1.0	1.8	0.2	1.3	0	0.3	0.4	1.1	0.5
11	1.7	1.7	0	0	0	0	0.7	0.9	0.8
12	1.3	1.7	0.5	0	0	0	0.1	0.8	0.2
13	1.0	1.5	0.5	0.2	0	0.3	0.3	0.5	0.2
14	0.2	0.8	0.3	0.3	0	0	0	0.3	0.2
15	0.5	1.5	0.2	0	0	0	0.1	0.2	0
16	0.3	0.3	0	0	0	0	0.1	0	0
17	0	0	0	0	0	0	0	0.2	0
18	0.3	0	0	0	0	0	0	0	0.2
19	0.2	0	0	0	0	0	0	0	0
20	0.3	0	0	0	0	0	0.1	0	0

## Data Availability

Figures and tables in this article show raw data or means combined with min/max and 25/75% of data box plots, hence visualizing the range of data comprehensively; however, raw data can be made available upon request if required.

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
