# Peer review of "Method for Isolation of Myxozoan Proliferative Stages from Fish at High Yield and Purity: An Essential Prerequisite for In Vitro, In Vivo and Genomics-Based Research Developments"

_cells, 2022, doi:10.3390/cells11030377_

Round 1

Reviewer 1 Report

In this manuscript, the authors present a new method to isolate myxozoan developmental stages from blood cells of host fish. Using DEAE-cellulose ion exchange chromatography, they were able to isolate and purify (99-100%) viable Sphaerospora molnari presporogonic stage cells based on their surface charge. The method is well structured, detailed and clearly described. Additionally, the authors present a rapid flow cytometric method for the quantification of parasite cells in full blood or in different separation fractions. Isolation of clean samples of parasite cells from host cells is highly valuable for omics analysis as well as for in vitro and in vivo experiments. This methodological manuscript is therefore important and relevant to the scientific community. The manuscript is well written and I have only a few minor comments.

Line 311-313- It is unclear to me why at the second elution with 1.2x RPMIsalt there was no host contaminant, whereas using first this solution “demonstrated a minor contamination with erythrocytes”.

Fig 2b- Please add a scale bar. In addition, it would be more informative to show light microscopy or DIC images near the DAPI staining, in order to display cellular boundaries.

Line 363- Fig 3- Please add a and b sections in the legend and in the figure. Is it correct that the red dots at the bottom resemble two nuclei/cells? Are there no single cells?

Supplementary Figures:

Fig S1 – Please explain more clearly the first panel in a. Is it a control without the anti-carp ab and only with the second ab?

Fig S2- Please add titles for Steps 1, 2, 3, 4 above the figure panels.

Fig S3- I could not understand the legend. What represents the measurements of the flow cytometry and what of the microscopy?

Author Response

We would like to thank the reviewer for the creative criticism that has helped to improve the clarity of the paper. Please find point-by-point responses to the comments provided:

Line 311-313- It is unclear to me why at the second elution with 1.2x RPMIsalt there was no host contaminant, whereas using first this solution “demonstrated a minor contamination with erythrocytes”.

It is possible that together with the host cell, a percentage of parasites are retained in the 1x RPMIsalt column by adhesion to the cellulose fibres. We assume that the slightly increased salinity of the 1.2x RPMIsalt liberates the less negatively charged parasite stages first when added to the 1x RPMIsalt column. We assume that with a gradual increase to 1.2x buffer the charge difference between parasite and host cells still favours their separation in the first 30 ml of buffer passing through. We did not continue elution beyond this stage.

Fig 2b- Please add a scale bar. In addition, it would be more informative to show light microscopy or DIC images near the DAPI staining, in order to display cellular boundaries.

Scale bar was added. Cell borders between individual cells of a single parasite stage are not visible unless the cytoplasm or cell membrane is stained. Hence, light microscopic/DIC images of DAPI (nuclear stain)-stained nuclei cannot visualize this. However, in this case the number of nuclei were essential for the study as we aimed to determine if the parasite stages were breaking up. Please find the general cell-in-cell characteristics of S. molnari blood stages visualised elsewhere (Lom et al. 1983, doi: doi:10.1017/S003118200005071X; Korytar et al. 2019, doi: 10.1186/S13071-019-3462-3)

Line 363- Fig 3- Please add a and b sections in the legend and in the figure. Is it correct that the red dots at the bottom resemble two nuclei/cells? Are there no single cells?

The figure was split into a and b sections and the respective legend formatted adequately. We did not observe single cell stages during the development of S. molnari in the blood, the smallest stage detected was a cell doublet composed of two cells.

Supplementary Figures:

Fig S1 – Please explain more clearly the first panel in a. Is it a control without the anti-carp ab and only with the second ab?

Correct. We clarified this in the legend.

Fig S2- Please add titles for Steps 1, 2, 3, 4 above the figure panels.

Titles were added.

Fig S3- I could not understand the legend. What represents the measurements of the flow cytometry and what of the microscopy?

The x axis shows the number of BS as determined by light microscopy while the y axis shows the number of BS as determined by flow cytometry (in the same sample; axes labelled accordingly in figure). If the two numbers are identical in different isolates then the dots would plot along the 45 degree line shown in the figure. In fact only one measurement is slightly off this line. This indicates good correlation between the data obtained by the two methods.

Reviewer 2 Report

Manuscript describes myxozoan proliferative stages purification by DEAE-C

The average size of a myxosporean spore usually ranges from 10 μm to 20 μm, whereas that of a malacosporean (a subclade of the Myxozoa) spore can be up to 2 mm. DEAE Columns have the 5-100 µm (C-grade?; not mentioned anywhere in the script) particle size and authors claim to be purifying even larger proliferative stages (multi cell packets) which are not necessarily spherical and identical making elution’s high loss and cause sampling bias for any downstream process owing to fractionating a subpopulation similar to each other .

Minor comments

The language needs to be revised for especially missing articles.

Review of literature is completely dissociated from the protocols. The choice of method is not defended by a sound previous report set.

Major comment

Authors claims are contradictory to the methods. Why cold standard methods were not used to compare with ion exchanger?

There is no test for live cells due to tough structure even dead cells can have intact nuclei. Same goes with antibody staining/painting

Why qPCR was not done with stage specific markers to honor the claim of live parasite as well as sample representing multiple stages.

Authors have themselves reported longer columns lead to more parasite loss. Isn’t this a indication of unsuitable method for the multistage population authors intend to isolate?

If authors developed pan-stage antibodies why they didn’t do preparative FACS directly to isolate pure parasites that too quickly.

The infectious potential of “purified” parasite has not been determined yet it is part of the claims placed in the abstract.

Author Response

We would like to thank the reviewer for the creative criticism that has helped to improve the clarity of the paper. Please find point-by-point responses to the comments provided:

The average size of a myxosporean spore usually ranges from 10 μm to 20 μm, whereas that of a malacosporean (a subclade of the Myxozoa) spore can be up to 2 mm. DEAE Columns have the 5-100 µm (C-grade?; not mentioned anywhere in the script) particle size and authors claim to be purifying even larger proliferative stages (multi cell packets) which are not necessarily spherical and identical making elution’s high loss and cause sampling bias for any downstream process owing to fractionating a subpopulation similar to each other.

It is specified in the text that blood stages of S. molnari contain 2-20 cells (lines 331 and 521) and measure 2.4-12 μm in diameter (line 432). Parasite stages are hence as small or smaller than fish blood cells. Parasites are spherical or oval as seen in Figure 2b and previously described (Korytar et al. 2019, doi: 10.1186/S13071-019-3462-3). Line 181 informs about the type of cellulose used, however, while the fibre size of the specific cellulose type is not specified by the producer on the product packaging or online, it is irrelevant to the separation, since separation is by surface charge and adhesion-elution rather than by a filtration-type retention in which size plays a role.

The language needs to be revised for especially missing articles.

The language has been revised once more and missing article added. I would like to point out that the author list contains one native Canadian (Dr. Chan) and one native US American (Dr. Panicucchi), i.e. 2 native English speakers, which have extensively revised the manuscript.

Review of literature is completely dissociated from the protocols. The choice of method is not defended by a sound previous report set.

We clearly specify the different methods that can be used for separation of cells from a heterogenous population (line 91 ff), we justify the choice of the adhesion-elution method chosen here (line 97 ff) and describe the literature related to this method (line 104 ff) as well as other methods previously applied in myxozoans (line 65 ff). In principle, the whole introduction of the paper as well as some sections in the discussion refer only to the methodology and the protocols used for cell separation. I don’t see how to what extent this should be enlarged any further.

Authors claims are contradictory to the methods. Why cold standard methods were not used to compare with ion exchanger?

Cold standard methods for myxozoan proliferative stages do not exist. However, we used our laboratory standard method based on density centrifugation (using Ficoll) for comparison throughout the study and showed that a higher number of parasites can be isolated at the cost of extensive host cell contamination (see Fig. 1 and results section L282 ff). We ensured optimization of the ion exchange protocol by testing a number of buffers of varying composition and ionic strengths as well as different column heights. We feel we fulfil the claim of performing a comparative analysis.

There is no test for live cells due to tough structure even dead cells can have intact nuclei. Same goes with antibody staining/painting

Nuclear staining was only used to determine the cellularity of parasite stages. Viability was tested using a live-dead staining method employing propidium iodide and fluorescein diacetate (protocol specified in L208 ff) and tested infectivity by injecting all isolates into fish as stated in L224. We found mortality to be low, ranging between 0% and 6% of all parasites (L328 ff) and all fish became infected after 4-6 weeks (stated in L349 ff).

Why qPCR was not done with stage specific markers to honor the claim of live parasite as well as sample representing multiple stages.

qPCR does not confirm the viability of the parasites, it merely identifies the presence of DNA in a sample which can persist also in dead organisms. The methods specified above are accepted methods to test for viability and further development and proliferation in fish is convincing proof of infectivity.

Authors have themselves reported longer columns lead to more parasite loss. Isn’t this a indication of unsuitable method for the multistage population authors intend to isolate?

Longer columns simply cause more unwanted parasite adhesion than shorter ones. We have demonstrated that the population of parasites that is eluted from the DEAE-C column shows a similar cellularity distribution as detected in whole blood (Fig. 2A), indicating that stages of different size (2-20 cells) are successfully eluted unharmed/without rupture.

If authors developed pan-stage antibodies why they didn’t do preparative FACS directly to isolate pure parasites that too quickly.

Good point. We tried to generate polyclonal antibodies directly against the parasite but since S. molnari feeds on host red blood cells (Korytar et al. 2020, doi:10.1111/PIM.12683) such parasite isolates contain also host proteins. Injection of the isolated parasites into mice resulted in an undesirable fraction of antibodies binding to host proteins. Additionally, we believe only a negative labelling approach allows recovery of an entirely intact parasite population for downstream analyses.

The infectious potential of “purified” parasite has not been determined yet it is part of the claims placed in the abstract.

As stated, DEAE-C isolated parasite populations were injected into SPF carp and all became infected after 4-6 weeks (L349) proving the infectivity of the isolates.

Reviewer 3 Report

The experiments described in this manuscript are well designed, and the paper is well written and easy to follow. The authors have done a great job of making their procedures reproducible and accessible to anyone interested in using this method in the future. This will be very useful for anyone studying Myxozoa and different life stages of these parasites.

I have a few minor comments:

Title: Not sure it is necessary to include "and in silico" in the title. Seems like this was a keyword thrown into the title, but it is never mentioned once in the paper. Suggest removal.

Lines 43-45: This could be phrased more clearly. As written, it makes it sound like the issue is that the fish are parasitized when they reach the consumer, whereas I think the true issue is that there is not a method of treatment or vaccine that is considered safe for later human consumption.

Line 80: maybe "blood is centrifuged"?

Line 125: change "are" to "were".

Lines 134-135: define the 3Rs of animal welfare in parentheses?

Line 145: define PBS or clearly state that it is a medium. It could get confused with BS which is used a lot prior to this line.

Line 154: delete "=".

Lines 207 and 230: italicize S. molnari.

Lines 278-279: Suggest a more general subheading title such as "Optimal S. molnari BS recovery".

Lines 303-304: The statement regarding what the data illustrates should be removed from the figure caption as it is already stated in the results section that refers to this figure.

Lines 314-315: Suggest a more general subheading title such as "Vitality, integrity and infectivity of S. molnari BS post-isolation".

Lines 353-354: Suggest a more general subheading title such as "Parasite elution" or "Parasite elution by fraction".

Lines 370-371: Suggest removing "in a fast, semi-automated manner" from subheading.

Line 419: such "as" myxosporean spores?

Line 493: delete "e.g.".

Lines 516-526: The conclusions are very repetitive of everything just stated in the discussion section and don't add much to the paper. I suggest deletion of this section. If a conclusion paragraph is desired, I suggest making the paragraph in lines 501-514 the conclusions paragraph as the results and implications are well summarized there.

Author Response

We would like to thank the reviewer for the creative criticism that has helped to improve the clarity of the paper. Please find point-by-point responses to the comments provided:

Title: Not sure it is necessary to include "and in silico" in the title. Seems like this was a keyword thrown into the title, but it is never mentioned once in the paper. Suggest removal.

We exchanged the term ‘in silico’ for ‘genomics-based’ as we feel we have explained the need for host contamination-free DNA and RNA sequencing data for organisms with strongly derived genomes several times (lines 55 and 536).

Lines 43-45: This could be phrased more clearly. As written, it makes it sound like the issue is that the fish are parasitized when they reach the consumer, whereas I think the true issue is that there is not a method of treatment or vaccine that is considered safe for later human consumption.

Sentence was rephrased.

Line 80: maybe "blood is centrifuged"?

Change was implemented as suggested.

Line 125: change "are" to "were".

Change was implemented as suggested.

Lines 134-135: define the 3Rs of animal welfare in parentheses?

Change was implemented as suggested.

Line 145: define PBS or clearly state that it is a medium. It could get confused with BS which is used a lot prior to this line.

Change was implemented as suggested.

Line 154: delete "=".

Change was implemented as suggested.

Lines 207 and 230: italicize S. molnari.

Change was implemented as suggested.

Lines 278-279: Suggest a more general subheading title such as "Optimal S. molnari BS recovery".

Change was implemented as suggested.

Lines 303-304: The statement regarding what the data illustrates should be removed from the figure caption as it is already stated in the results section that refers to this figure.

Change was implemented as suggested.

Lines 314-315: Suggest a more general subheading title such as "Vitality, integrity and infectivity of S. molnari BS post-isolation".

Change was implemented as suggested.

Lines 353-354: Suggest a more general subheading title such as "Parasite elution" or "Parasite elution by fraction".

Change was implemented as suggested.

Lines 370-371: Suggest removing "in a fast, semi-automated manner" from subheading.

Change was implemented as suggested.

Line 419: such "as" myxosporean spores?

Mistake was corrected.

Line 493: delete "e.g.".

Deleted.

Lines 516-526: The conclusions are very repetitive of everything just stated in the discussion section and don't add much to the paper. I suggest deletion of this section. If a conclusion paragraph is desired, I suggest making the paragraph in lines 501-514 the conclusions paragraph as the results and implications are well summarized there.

Change was performed as suggested.

Round 2

Reviewer 2 Report

It's hard to believe that I read the following in a rebuttal...

" while the fiber size of the specific cellulose type is not specified by the producer on the product packaging or online, it is irrelevant to the separation, since separation is by surface charge and adhesion-elution rather than by filtration-type retention in which size plays a role."

This response is a red flag as makes the "groundbreaking" method not reproducible in any other laboratory. Even not reproducible in the same lab if they do not buy online from the same company that doesn't provide pore size... And to write that size does not play a role in Ion exchanger especially the particle size of the resin makes me wish this was written by one of the authors and overlooked by other renowned members of the team. 

The novelty of this project is to have a breakthrough in purifying parasites, which is in question due to the lack of technical details making the proposed method irreproducible. Also, rebuttal boldly claims to have information regarding non-specificity of the antibodies which is missing from the paper. 

Author Response

We hereby reply to the reviewer's comments and we further implemented all required changes into the manuscript.

Reviewer's comment:  It's hard to believe that I read the following in a rebuttal...

" while the fiber size of the specific cellulose type is not specified by the producer on the product packaging or online, it is irrelevant to the separation, since separation is by surface charge and adhesion-elution rather than by filtration-type retention in which size plays a role."

This response is a red flag as makes the "groundbreaking" method not reproducible in any other laboratory. Even not reproducible in the same lab if they do not buy online from the same company that doesn't provide pore size... And to write that size does not play a role in Ion exchanger especially the particle size of the resin makes me wish this was written by one of the authors and overlooked by other renowned members of the team. 

The novelty of this project is to have a breakthrough in purifying parasites, which is in question due to the lack of technical details making the proposed method irreproducible. 

Answer to reviewer: We inquired about the details of the cellulose type used in our experiments and provide all information provided by the company (manuscript L174). These include fibre/granule size, exchange capacity, water content, flow rate and protein loading capacity. Cellulose fibres/granules lack pores so we cannot report pore size. We hope that inclusion of the detailed information on the product will satisfy the reviewer.

Reviewer's comment: Also, rebuttal boldly claims to have information regarding non-specificity of the antibodies which is missing from the paper. 

The reviewer is correct about how non-specific binding of our pan-carp cell-specific reagent to the parasite may factor into inaccurate measurements. It is true that, although we demonstrate that our reagent can stain nearly all blood-circulating host cells of the carp (please, refer to Figure S1a), the application of the reagent may be imperfect due to a) the nature of the reagent (a polyclonal antibody) and b) the nature of the target (a myxozoan parasite that can incorporate carp host proteins onto its cell surface). However, it is either manageable or negligible.   a) We developed a polyclonal reagent that ideally should bind to everything except the parasite, such that the latter can be counted and studied. The opposite approach is not yet possible and prone to disturbing parasite viability. We have yet to identify a parasite surface protein shared by all stages and even if we did, a monoclonal antibody targeting such a hypothetical target is prone to crosslinking the target which can trigger undesired signaling events. To achieve our goal, we immunized mice with carp blood cells to raise antibodies that are as clonally diverse as the heterogeneity of their targets. The caveat is that the peptide sequence diversity of both the antibodies and their targets increase the likelihood of non-specific protein-protein interactions. By chance, the peptide or epitope of a host protein can match or be similar to that of a parasite protein.   b) This problem is exacerbated by the fact that the parasite can incorporate host red blood cell (the most abundant cell type in circulation) surface proteins as part of its life cycle and potentially for evading the host immune system (please, refer to https://doi.org/10.1111/pim.12683). Thus, a certain portion of the polyclonal antibodies may theoretically bind the parasite, especially larger stages.

However, we demonstrated that the reagent is capable of staining virtually all blood cells for complete separation from cells to which the reagent is not added (Figure S1a). More antibody binding translates to more fluorescence emitted from labeled antibodies and points (representing single cells) that shift up in proportion to increased staining intensity (Y-axes). Ideally, the parasite should react much like non-stained host cells (Figure S1a, left panel) whereas host cells can be easily distinguished from non-stained cells (Figure S1a, right panel). However, because of the nature of the reagent and the nature of the target, there is a slight discernible shift when comparing non-stained parasite to stained parasite (Figure S1b, compare left to right panel). However, this non-specific binding is negligible because the parasite population does not shift to the same degree as host cells (Figure S1a) and the few parasite cells that are misidentified into the box/limits we established are also negligible. The result is that we are able to distinguish host cells from the parasite (Figure 4), especially for smaller parasite stages, and that flow cytometry results correlate with those of microscopy (Figure S3).

Please also see additional text regarding antibody specificity implemented into L465-487 of the manuscript.